# SAA-UNet: Spatial Attention and Attention Gate UNet for COVID-19 Pneumonia Segmentation from Computed Tomography

**DOI:** 10.3390/diagnostics13091658

**Published:** 2023-05-08

**Authors:** Shroog Alshomrani, Muhammad Arif, Mohammed A. Al Ghamdi

**Affiliations:** Department of Computer Science, Umm Al-Qura University, Makkah 24382, Saudi Arabia

**Keywords:** COVID-19 pneumonia segmentation, CT images, SAA-UNet model, spatial attention module (SAM), attention gate (AG)

## Abstract

The disaster of the COVID-19 pandemic has claimed numerous lives and wreaked havoc on the entire world due to its transmissible nature. One of the complications of COVID-19 is pneumonia. Different radiography methods, particularly computed tomography (CT), have shown outstanding performance in effectively diagnosing pneumonia. In this paper, we propose a spatial attention and attention gate UNet model (SAA-UNet) inspired by spatial attention UNet (SA-UNet) and attention UNet (Att-UNet) to deal with the problem of infection segmentation in the lungs. The proposed method was applied to the MedSeg, Radiopaedia 9P, combination of MedSeg and Radiopaedia 9P, and Zenodo 20P datasets. The proposed method showed good infection segmentation results (two classes: infection and background) with an average Dice similarity coefficient of 0.85, 0.94, 0.91, and 0.93 and a mean intersection over union (IOU) of 0.78, 0.90, 0.86, and 0.87, respectively, on the four datasets mentioned above. Moreover, it also performed well in multi-class segmentation with average Dice similarity coefficients of 0.693, 0.89, 0.87, and 0.93 and IOU scores of 0.68, 0.87, 0.78, and 0.89 on the four datasets, respectively. Classification accuracies of more than 97% were achieved for all four datasets. The F1-scores for the MedSeg, Radiopaedia P9, combination of MedSeg and Radiopaedia P9, and Zenodo 20P datasets were 0.865, 0.943, 0.917, and 0.926, respectively, for the binary classification. For multi-class classification, accuracies of more than 96% were achieved on all four datasets. The experimental results showed that the framework proposed can effectively and efficiently segment COVID-19 infection on CT images with different contrast and utilize this to aid in diagnosing and treating pneumonia caused by COVID-19.

## 1. Introduction

In December 2019, people began rush Wuhan hospitals with severe pneumonia of unknown cause. After the number of infected people increased, on 31 December, China notified the World Health Organization of the outbreak [1,2]. After several examinations, the virus was found to be a coronavirus with more than 70% similarity to SARS-CoV on 7 January [3]. Coronavirus 2019 is a severe acute respiratory syndrome (SARS-CoV-2), named COVID-19 by the World Health Organization in February 2020 [4]. It is from the beta virus family, which is highly contagious and causes various diseases. One of these viruses appeared in 2003, called severe acute respiratory syndrome (SARS), and another appeared in 2012, the Middle East respiratory syndrome (MERS) [5,6]. The first fatal case of coronavirus was reported on 11 January 2020. As a result, the World Health Organization (WHO) declared a global emergency on 30 January 2020. The number of cases began to increase dramatically due to human-to-human transmission [7]. The infection is transmitted through droplets from the coughing and sneezing by patients, whether they show symptoms or not [8]. These infected droplets can spread from one to two meters and accumulate on surfaces. COVID-19 continued to spread despite strict preventive efforts. Consequently, the WHO declared coronavirus a global pandemic at the International Health Meeting held in March 2020 [9]. The number of confirmed cases has reached more than 758 million, and the number of deaths has reached 6,859,093 persons [10].

Pneumonia is a complication of viral diseases such as COVID-19, influenza, the common cold, bacteria, fungi, and other microorganisms. COVID-19 can affect any organ in the human body, and the symptoms range from mild, like the common cold, to more severe pneumonia or even be asymptomatic. Pneumonia caused by COVID-19 is named “novel coronavirus-infected pneumonia (NCIP)” [11].

The formal diagnosis of COVID-19 infection is the reverse-transcription-polymerase chain reaction (RT-PCR) test. This test takes a swab from the mouth, nasopharynx, bronchial lavage, or tracheal aspirate. The RT-PCR test has a high error rate because of the low sensitivity. Furthermore, blood tests may show signs of COVID-19 pneumonia [12]. Computed tomography (CT) of the chest is a complementary tool for the diagnosis even before the patients develop symptoms, as CT images show the places of lung damage caused by COVID-19 [13]. This helps to know the extent of the infection at any stage of the disease. CT is the latest tool that uses X-rays and computers to create three-dimensional human body images. It is a scan that combines a series of X-ray images taken from different angles around an organ or body and uses computer processing to create cross-sectional images called slices. Computerized tomography images provide more detailed information than regular X-rays, as they are three-dimensional images. These 3D images are made using tomography, which shows the parts of the organ, facilitates segmentation, and diagnoses diseases. In CT scans for people with COVID-19, the lungs contain different opacity forms such as ground-glass opacity (GGO) and consolidation, as shown in Figure 1 [14]. This infection is due to the entry of the virus into the cells by attaching to surface angiotensin-converting receptor enzyme 2 (ACE2). After the virus enters, it causes the tiny air sacs to inflate, causing them to fill with so much fluid and pus that breathing is difficult. The inhaled oxygen is processed and delivered to the blood in these sacs. This damage causes tissue rupture and blockage in the lungs. Later, the walls of these sacs thicken, making breathing difficult. As a result of that, the lungs become the first organ affected by the coronavirus [15,16].

Artificial intelligence, specifically deep learning, has recently played an effective and influential role in medical images. The diagnostic evaluation of medical image data is a human-based technique that requires sufficient time by expert radiologists. Recent advances in artificial intelligence have substituted many personalized diagnostic procedures with computer-aided diagnostic (CAD) methods that can achieve effective real-time diagnoses. As a result, it has an essential role in diagnosing diseases such as infections, cancer, and many other diseases by taking shots of the organ or even the whole body to help radiologists make decisions and plan the stage of treatment. The segmentation task identifies the pixel or voxels that make up the contour or the interior of the region of interest (ROI) as the first stage in computer-aided diagnostics (CAD) [17,18]. Many deep learning algorithms used in image segmentation tasks have succeeded in biomedical images. For example, a fully convolutional network (FCN) was proposed as an end-to-end, pixel-to-pixel network for image segmentation [19], SegNet [20]. UNet was proposed for biomedical image segmentation, in which an encoder–decoder structure with concatenated skip connections yielded significant performance improvements [21], and the modified UNet (UNet++ [22]) and PSPnet [23] have been widely used in medical image segmentation.

This research proposes a spatial attention and attention gate UNet model (SAA-UNet). Additionally, we trained the SAA-UNet model with boundary loss combined with weighted category cross-entropy and Dice loss as a loss function. The framework was used to identify areas of COVID-19 pneumonia and segment regions of interest (ROIs) from computed tomography images. We applied it to four limited datasets published in open sources at the European Institute for Biomedical Imaging Research (EIBIR) [24]. The summary of the contributions of this work is as follows:We propose the spatial attention and attention gate UNet model (SAA-UNet) based on attention UNet (Att-UNet) and spatial attention UNet (SA-UNet). We took the attention approach proposed by Ozan Oktay et al. [25] to focus on COVID-19 infection regions. The local features vector of infection improved the performance compared to gating established on a global feature vector. We took the spatial attention module (SAM) approach proposed by Changlu Guo et al. [26] to deal with features fed to the bridge of SAA-UNet from the encoder to the decoder. This makes it take essential features needed in spatial information and helps reduce the number of parameters.SAA-UNet proved to be effective in segmenting the infection areas in CT images of COVID-19 patients.SAA-UNet showed good generalization when applied to different datasets.

The paper is organized as follows: Section 2 provides the related literature review. Section 3 describes the proposed framework, spatial attention, and attention gate UNet (SAA-UNet) model architecture in detail. Section 4 describes the COVID-19 CT image datasets, and Section 5 explains the analysis and preprocessing of the data. Section 6 shows the experimental results, and Section 7 provides the discussion on the experimental results. Finally, Section 8 concludes the paper and provides future work recommendations.

## 2. Related Work

With artificial intelligence (AI) advancements in the health field, many deep learning algorithms have been proposed for medical image processing as segmentation tasks play an essential role in the treatment stage. For example, Ronneberger et al. [21] introduced the standard UNet for biomedical image segmentation. They evaluated UNet on several datasets, including the ISBI Challenge for segmenting neuronal structures in electron microscopic stacks. They achieved an average IOU on the PhC-U373 dataset of 0.92 and on DIC-HeLa of 0.777. Oktay et al. [25] proposed an extension to the UNet architecture. They added an attention mechanism to skip the connection of UNet to focus on the image’s region of interest and improve the segmentation. They evaluated attention UNet on the 150 abdominal 3D CT scans from patients diagnosed with gastric cancer dataset and achieved a Dice score of 0.84. The second dataset CT consisting of 82 contrast-enhanced 3D CT scans of the pancreas achieved a Dice score of 0.831. In continuation, Zhao et al. [26] proposed a modification of the UNet architecture that included a spatial attention module in the bridge to focus on the important regions of the image. They evaluated SA-UNet on the Vascular Extraction (DRIVE) dataset and the Child Heart and Health Study (CHASE-DB1) dataset. They achieved F1-scores of 0.826 and 0.815, respectively.

Relying on the above, deep learning models can be used to find areas of lung damage caused by 2019-nCoV. Athanasios Voulodimos et al. [27] used an FCN-8s to segment COVID-19 pneumonia and achieved a 0.57 Dice coefficient. They proposed a light UNet model with three stages of the encoder and decoder to deal with the limited datasets of this problem. This achieved a 0.64 Dice coefficient. Sanika Walvekar and Swati Shinde proposed UNet with preprocessing and spatial, color, and noise data augmentation from the MIScnn library with Tversky loss [28]. The Dice similarity coefficient (DSC) for COVID-19 was 0.87 for infection segmentation and 0.89 for the lungs. Imran Ahmed et al. [29] proposed an attention mechanism added to the standard UNet architecture to improve feature representation with binary cross-entropy Dice loss and boundary loss. The Dice score was 0.764 on the validation set. Tongxue Zhou et al. [30] proposed a spatial attention module and a channel attention module added to a UNet architecture with focal Tversky loss. The spatial attention module reweights the feature representation spatially and channelwise to capture rich contextual relationships for better feature representation. The DSC was 0.831. Narges Saeedizadeh et al. [31] proposed a ground-glass recognition system called TV-Unet, a UNet model with a total variation gradient. The loss function was the binary cross-entropy with a total variation term. The DSC achieved 0.86 and 0.76 for two different splits. The combination of two UNet models proposed by Narinder Singh Punna and Sonali Agarwala [31] is called the CHS-NET model. One segments the lungs, and the other segments infection with the weighted binary cross-entropy and Dice loss function. The CHS-NET model uses UNet, Google’s Inception model, a residual network, and an attention strategy. The DSC for the lungs was 0.96, whereas for COVID-19 infection, it was 0.81. Tal Ben-Haim et al. [32] proposed a VGG backbone in the encoder of two UNets. The first UNet model segments the lung regions from CT images. The second UNet model extracts the infection or shapes of lesions (GGO and consolidation). For the segmentation of infection with the binary cross-entropy loss, the DSC was 0.80, and for the multi-class weighted cross-entropy (WCE) and Dice loss, the GGO was 0.79 DSC and the consolidation 0.68. A plug-and-play attention module [33] was proposed to extract spatial features by adding to the UNet output. The plug-and-play attention module contains a position offset to build the positional relationship between pixels. This framework achieved 0.839 for the DSC. Ziyang Wang and Irina Voiculescu [34] proposed the quadruple augmented pyramid network (QAP-Net) for multi-class segmentation by establishing four augmented pyramid networks on the encoder–decoder network. These four were two pyramid atrous networks with different dilation rates, the pyramid avg pooling network and the pyramid max pooling network. The mean intersection over union (IOU) score with categorical focal loss was 0.816. Qi Yang et al. [35] used MultiResUNet [36] as the basic model, introduced a new “Residual block” structure in the encoder part, added regularization and dropout, and changed the partial activation function from rectified linear unit (ReLU) activation function to LeakyReLU. The DSC with a combination of binary cross-entropy, focal, and Tversky loss was 0.884. Nastaran Enshaei et al. [37] proposed using the Inception-V3, Xception, InceptionResNet-V2, and DenseNet-121 pre-trained encoders and replacing each fully connected model with the decoder to segment COVID-19 infection. Consequently, the the results of multiple models were aggregated by soft voting for each image pixel. This achieved a Dice score for GGO = 0.627 and consolidation = 0.592 with the categorical cross-entropy. Moreover, Murat Ucar [38] proposed aggregating the pre-trained VGG16, ResNet101, DenseNet121, InceptionV3, and EfficientNetB5 with a pixel-level majority vote to obtain the last class probabilities for each pixel in the image. The Dice coefficient was 0.85 with the Dice loss. Hong-Yang PEI et al. [39] proposed a multi-point supervised network (MPS-Net) based on UNet. The proposed model gave a 0.833 DSC result with a combination of binary cross-entropy and Tversky loss to detect COVID-19 infection. Ümit Budak et al. [40] proposed an A-SegNet network that combines SegNet with the attention gate (AG) mechanism. The DSC score was 0.896 on the validation set with focal Tversky loss.

Alex Noel Joseph Raj et al. proposed an attention gate-dense network-improved dilation convolution UNet (ADID-UNET) based on UNet [41]. ADID-UNet achieved an average Dice score of 0.803 on the MedSeg + Radiopaedia dataset with the Dice loss. Ying Chen et al. proposed a HADCNet model based on UNet that contains hybrid attention modules in five stages of the encoder and decoder [42]. It helps balance the semantic differences between various levels of features, which refines the feature information. HADCNet was trained with five-fold cross-validation with the cross-entropy and Dice loss on the MedSeg, Radiopaedia P9, 150 COVID-19 patients, and Zenodo datasets, achieving Dice scores of 0.792, 0.796, 0.785, and 0.723. Nour Eldeen M. Khalifa et al. proposed an architecture of three encoder and decoder stages to deal with the limited datasets problems [43]. The mean IOU score for Zonodo 20P achieved 0.799. Yu Qiu et al. proposed a MiniSeg model to extract multiscale features and deal with limited datasets with 83K parameters [44]. After MiniSeg was trained with five-fold cross-validation with the cross-entropy loss on MedSeg, Radiopaedia (P9), Zenodo 20P, and MosMedData, the average Dice scores were 0.759, 0.80, 0.763, and 0.64, respectively. Xiaoxin Wu et al. proposed a focal attention module (FAM) inspired by a residual attention network that contains channel and spatial attention, with a residual branch in the feature map [45]. The focal attention module was applied to the FCN, UNet, SegNet, PSPNet, UNet++, and DeepLabV3+ with binary cross-entropy loss (BCE), where the best was DeepLabV3+ when applied on Zenodo 20P with an average Dice score of 0.885. Feng Xie et al. proposed the double-U-shaped dilated attention network (DUDA-Net) to enhance segmentation [46]. DUDA-Net contains a coarse-to-fine network with a coarse network for lung segmentation and a fine network for infection segmentation. The proposed model was trained with five-fold cross-validation with Tversky loss on infection slices of Radiopaedia 9P with an average Dice score of 0.871 and a mean IOU of 0.771. Vivek Kumar Singh et al. proposed a LungInfseg model based on an encoder and decoder structure [47]. LungInfseg was applied on Zenodo 20P with a combination of blockwise (BWL) and total loss (TL), with an average Dice score of 0.8034. R. Karthik et al. proposed a contour-enhanced attention decoder CNN model with an encoder and decoder structure [48]. The proposed model with the mean pixelwise cross-entropy loss was applied to the Zenodo 20P dataset and had an average Dice score of 0.88; on the MosMedData dataset, the Dice score was 0.837, and on the combination of the Zenodo 20P and MosMedData datasets, the Dice score was 0.854. Kumar T. Rajamani et al. proposed the deformable attention net (DDANet) model [49] based on UNet and criss-cross attention (CCNet) [50]. The proposed model has the same structure as attention UNet [25], with a criss-cross attention module inserted in the bottleneck to capture non-local interactions. DDANet was trained with five-fold cross-validation on the combined dataset of MedSeg and Radiopaedia 9P with multiple classes with class-weighted cross-entropy loss where GGO was 0.734, consolidation was 0.614, and the average Dice score was 0.781.

Three-dimensional algorithms can be used for the overall CT volume of a patient. Keno K. Bressem [51] proposed a pre-trained 3D ResNet block added to the 3D UNet architecture to solve COVID-19 computed tomography image segmentation. The DSC was 0.648, combining the Dice loss and pixelwise cross-entropy loss. Aswathy A. L. and Vinod Chandra [52] proposed a cascaded 3D UNet with two 3D UNet, the first for segment lung volumes and the second for infection volume. The DSC for the lung = 0.925 and infection = 0.82. The 3D algorithms for the segmentation of COVID-19 from CT are rarely used for several reasons, including the computational cost and limited datasets of this problem.

This research proposes a framework to train SAA-UNet in binary and multi-class segmentation using the contrast enhancement method in preprocessing and a combination of the weighted category cross-entropy, Dice, and boundary loss as the loss function. The boundary loss function with regional loss takes useful information from infection bounds from irregular and complex shapes.

## 3. Methodology

The spatial attention and attention mechanism UNet model (SAA-UNet) is a proposed state-of-the-art algorithm based on spatial attention UNet (SA-UNet) [26] and attention UNet (Att-UNet) [25] to deal with the complexity of COVID-19 pneumonia images. Moreover, a framework is proposed using the preprocessing method and a combination of weighted category cross-entropy, Dice loss, and boundary loss.

The flowchart of the framework to train the proposed model followed in this research is illustrated in Figure 2. At the beginning, the slices $x_{i}$ are extracted from the CT scan $x_{I}$ if the dataset is not initially of slice images. Afterward, $x_{i}$ are fed to the preprocessing phase, and then, the pixel is classified as either binary or multi-class. Then, the dataset is split into training and testing sets and the training set fed as the input to SAA-UNet to train with 10-fold cross-validation. Next, the trained model is tested on the test set. Finally, the masks of the images of the region of interest (ROI) of COVID-19 damage in the lungs are predicted.

The section is organized as follows: Section 3.1 is the CT preprocessing stage. Section 3.2 is the SAA-UNet model architecture’s description with the details of the spatial attention module (SAM) in Section 3.3 and the attention gate (AG) in Section 3.4. After that, Section 3.5 explains the optimization with the combination of the weighted category cross-entropy, Dice, and boundary loss. Finally, Section 3.6 displays the performance metrics used to evaluate the SAA-UNet model.

### 3.1. Pre-Processing of Images

The Hounsfield unit (HU) scale is a dimensionless unit utilized in CT images depending on the organ and the disease. The chest CT pixel value intensity of air is −1000, of water is 0, of the lung is −700 to −500, and of the lung tissue is 500 HU to 910 HU, whereas the chest wall, blood, and bone are higher than 500 HU. The HU is used due to the imperfect clarity of CT scan datasets before entering them into the model. The CT scan contrast is different from one dataset to another. As shown in Figure 2, the preprocessing stage begins with the edit Hounsfield unit (HU) histogram if the intensity of $x_{i}$ pixels of air is less than −1000. This means that the datasets have insufficient contrast, so normalizing the air by more than −1000 allows the contrast to increase. The contrast stretching is enhanced when the helpful $x_{i}$ pixels on the left edge are mapped to black and the right ones to white. As a result, the useless pixels are removed by creating a threshold with two cutoff points (Equation 1).
(1)$$Threshold=\left(x_{i}>HU_{lower}\right)and\left(x_{i}<HU_{higher}\right)$$ These are generated by a Boolean mask from the NumPy array and selecting values between the lower and upper bounds [53]. After enhancing the contrast, $x_{i}$ is normalized and confined between 0 and 1. Then, $x_{i}$ is rotated with the related masks 90 degrees. The final step in the preprocessing is resizing different resolutions of $x_{i}$ by the OpenCV library [54] to decrease the cost compensation with inter-area interpolation, resampling using the pixel area relation.

### 3.2. Spatial Attention and Attention UNet Model

SAA-UNet has an encoder–decoder structure, as shown in Figure 3. The encoder phase has four stages: E1, E2, E3, and E4, which help extract the information from the CT slices’ input images. At the beginning, with binary segmentation, $x_{i}$ is fed as the input to E1, consisting of two convolutional layers with a 3 × 3 kernel size, stride 1, and 64 filters, each followed by the ReLU activation function (Equation 2), then a 2 × 2 Max-Pooling layer to progressively decrease the spatial size of the representation.
(2)$$ReLU:f\left(x\right)=Max\left\{0,x\right\}$$ E2, E3, and E4 consist of two 3 × 3 convolutional layers with 128, 256, and 512 filters, respectively, and stride 1. Each convolutional layer is followed by batch normalization, ReLU activation functions, and 2 × 2 Max-Pooling. Each output of Max-Pooling is fed to the next encoder stage. Consequently, the E4 output is fed into the bridge that contains the spatial attention module (SAM). The SAM helps extract the spatial features from all encoder stages and decreases the number of parameters. After that, the output of the SAM $F_{SAM}\in R^{H\times W\times 1}$ is fed to the decoder. Moreover, the extracted features map of each encoder stage are transferred by a skip connection to the corresponding decoder stage as a UNet model. The skip connection contains an attention gate (AG) to focus on essential features.

The decoder includes the D1, D2, D3, and D4 stages to determine the spatial information. Each stage has an upsample layer followed by a convolutional layer, batch normalization, and ReLU. The output of ReLU is forwarded to the attention gate (AG) as the second input. The output of the AG $F_{AG}\in R^{H\times W\times 1}$ is concatenated with the second input of the AG and goes to two convolutional layers and two ReLUs. The AGs filter the neuron activations to concentrate on a subset of target structures through the forward and backward passes. Through the backward pass, the gradients originating from background regions are down-weighted. This allows updating the model parameters in shallower layers based on relevant spatial regions. The AG parameters can be trained with the standard back-propagation updates. The D1 convolutional has 512 filters, whereas D2 has 256, D3 has 128 filters, and D4 has 64 filters, the same as the E1 filters. D4’s last layer is the 1 × 1 convolutional layer with a Sigmoid function for predicting binary masks.
(3)$$Sigmoid:f\left(x\right)=\frac{1}{1+e^{\left(-x\right)}}$$
where *x* is the input vector. In contrast, the multi-class segmentation of D4’s last layer has the 1 × 1 convolutional with the SoftMax function.
(4)$$SoftMax:\sigma {\left(x\right)}_{i}=\frac{e^{x_{i}}}{\sum_{j=1}^{k}e^{x_{j}}}$$
where $x_{j}$ is the input vector, $x_{i}$ is an element of the vector, and *k* is the number of classes. The multiple classes are learned with multi-dimensional attention coefficients, which have been used to learn sentence embedding [25]. Algorithm 1 explains the pseudocode of the SAA-UNet algorithm.

### 3.3. Spatial Attention Module

The spatial attention module is the informative part that focuses on producing a spatial attention map through the spatial association between features [26,55]. As illustrated in Figure 3, the output of the last E4 layer is fed as the input to the SAM. Figure 4 shown the input feature of SAM is $F\in R^{H\times W\times 1}$, which is forwarded through the channelwise Max-Pooling and Average-Pooling to generate the outputs $F_{Max}^{s}\in R^{H\times W\times 1}$ and $F_{Avg}^{s}\in R^{H\times W\times 1}$, respectively. These output feature maps are concatenated to make feature descriptors. Then, this is followed by the convolutional layer with a 7 × 7 kernel size and the Sigmoid activation function. After that, the output of the Sigmoid function layer is elementwise multiplication with E4 to generate a spatial attention map $F_{SAM}\in R^{H\times W\times 1}$.
(5)$$F_{SAM}=F\cdot \sigma (f^{7\times 7}\left(\left[F_{Max}^{s}\times F_{Avg}^{s}\right]\right)$$
where $f^{7\times 7}$ denotes a convolution operation with a kernel size of 7 and σ represents the Sigmoid function.

### 3.4. Attention Gating Module

The attention gate with additive attention focuses on capturing a sufficiently receptive feature map and identifies feature responses to keep only the relevant ones in the region of interest [25]. In this way, it progressively suppresses feature responses in irrelevant background regions without the necessity of cropping a region of interest (ROI). The AG is applied to the features, which are passed to the skip connection from the encoder stage, as shown in Figure 3, to disambiguate irrelevant and noisy responses. The two inputs to the AG are the corresponding encoder’s feature map and the decoder stage of deciding on the focus regions. As shown in Figure 5, each of the two inputs is fed to the 1 × 1 convolutional and batch normalization layers, and then, the two outputs are fed to the elementwise addition. After that, the output is fed to the ReLU activation, 1 × 1 convolutional, and batch normalization layers and the Sigmoid activation function. The output of the Sigmoid function is fed to the elementwise multiplication with the output of the last encoder stage layer.
(6)$$F_{AG}=F\cdot \sigma [\left(B\times f^{1\times 1}\times ReLU\right)\times (B\times f^{1\times 1})+\left(B\times f^{1\times 1}\right)]$$
where $f^{1\times 1}$ denotes a convolution operation with a kernel size of 1, B is batch normalization, and σ represents the Sigmoid function.
**Algorithm 1:** The pseudocode of the proposed SAA-UNet model  1 Image: input image of the network;  2 Encoder stages: E1, E2, E3, and E4, decoder stages: D1, D2, D3, and D4;  3 While (stage≤4), do  4  Converting image to feature map F;  5  For encoder stages ∈ {1, 2, 3, 4}, do  6   F → E1;  7   E1 → E2 and D4;  8   E2 → E3 and D3;  9   E3 → E4 and D2;10   E4 → SAM and D2;11  End for12  For decoder stages ∈ {1, 2, 3, 4}, do13   Generate spatial attention (SAM) by EQ(5);14   SAM → D1;15   D1 → D2;16   D2 → D3;17   D3 → D4;18  In D1, D2, D3, and D4: generate attention gate (AG) by EQ(6);19  End for20  Obtain the final feature map in binary segmentationwith EQ(3) and in multi-class segmentationby EQ(4);21 End while

### 3.5. Combination of Weighted Cross-Entropy Loss Function, Dice Loss, and Boundary Loss

When the segmentation model segments the infection from an organ, it will likely ignore small-sized anterior layers in the training process, resulting in low segmentation performance. In COVID-19 infection segmentation, the class imbalance problem can be solved using the loss function as an optimization method. In this study, a combination of the weighted cross-entropy loss function and Dice loss was used as the region loss function to combine their usefulness for the imbalanced dataset problem. Moreover, the boundary loss was integrated to take care of the edge information between regions and does not ignore them, like the other region losses.

Weighted cross-entropy loss is used to control category classification to calculate the probability of being a specific class, as proposed by Warren Weaver [56]. The basic formula is
(7)$$L_{CCE}=\frac{1}{n}\sum_{i=1}^{n}\sum_{j=1}^{m}y_{ij}\log\left(p_{ij}\right)$$
where *i* is the index of the samples, *j* is the index classes, *y* is the sample label, and $p_{ij}\in \left(0,1\right):\sum_{j}pij=1\forall i,j$ is the prediction for a sample. Moreover, *m* is the number of classes (in binary segmentation, *m* = 2, which is a special case of category cross-entropy called Bernoulli cross-entropy loss [57]).

Dice loss is inspired by the Dice score scale and is widely used in medical image segmentation to handle data imbalance problems. Nevertheless, it addresses the imbalance between foreground and background and between uncomplicated and complex examples that affect a learning model’s training process. It can be formulated as follows:(8)$$L_{Dice}\left(G,P\right)=1-\frac{2\left|P\cap G\right|}{\left|P\right|+\left|G\right|}$$
where *G* is the ground truth and *P* is predicted.

The combination of weighted category cross-entropy (*CCE*) loss and *Dice* loss as the region loss is given as
(9)$$Loss_{region}=wL_{CCE}+L_{Dice}$$
where *w* is the respective weight.

Boundary loss was proposed by Hoel Kervadec et al. [58,59], motivated by discrete optimization techniques for computing gradient flows of curve evolution. Boundary loss is a loss complimentary to region loss that integratesover the boundary instead of integratingover regions address the unbalanced segmentation problems. It is computed as the distance distribution $Dist(\partial G,\partial S_{\theta })$ between two boundaries in the spatial domain Ω, the *G* boundary ground truth of the spatial neighbor in the background Ω/G and $S_{\theta }$ the boundary segmentation region produced by the network.

The final boundary loss function is formulated as
(10)$$BL=\int_{\Omega }\phi_{G}\left(p\right)s_{\theta }\left(p\right)dp$$
where $\phi_{G}$ is pre-computed directly from the ground truth region *G*, $s_{\theta }\left(p\right)$ is the SoftMax probability outputs of the network with a constant independent of θ, and dp is independent of the network parameters.

Finally, the combination of the region loss (weighted category cross-entropy, Dice loss) with the boundary loss is formulated as
(11)$$Loss=\left(\alpha \right)Loss_{region}+\left(1-\alpha \right)BL$$
where α is a parameter balancing the losses. We started with a low value of α>0 and increased it gradually at the end of each epoch.

### 3.6. Performance Metrics

Four commonly used performance metrics in the field of medical image segmentation are the Dice coefficient score, the intersection over union (IoU) score, the sensitivity, and the specificity. We also computed the overall accuracy, precision, and F1-score to supplement the efficacy of the proposed model. The evaluation metrics, the accuracy, sensitivity, specificity, precision, and F1-score, were calculated based on the true positives (TPs), true negatives (TNs), false positives (FPs), and false negatives (FNs).

Pixel accuracy is the easiest way to evaluate the segmentation model’s performance.
(12)$$Pixel-Accuracy=\frac{TP+TN}{TP+TN+FP+FN}$$

Precision is a metric that measures the quality of predictions.
(13)$$Precision=\frac{TP}{TP+FP}$$
*Specificity* is also called the true negative rate (TNR) and measures the true negatives correctly determined by the model.
(14)$$Specificity=\frac{TN}{TN+FP}$$
*Sensitivity* (*recall*) is used to evaluate the model performance by showing how many positive instances the model correctly identified.
(15)$$Sensitivity\left(Recall\right)=\frac{TP}{TP+FN}$$ The *F*1-score is calculated by:(16)$$F1-score=2\times \frac{Precision\times Recall}{Precision+Recall}$$

The most-common measures to estimate segmentation are the Dice coefficient score and the intersection over union (IOU) score. The Dice coefficient score is two multiplications of the overlapping area between the ground truth and predicted segmentation divided by the total number of pixels in both images. It can be calculated as follows:(17)$$Dice\left(P,G\right)=\frac{2\left|P\cap G\right|}{\left|P\right|+\left|G\right|}=\frac{2TP}{2TP+FN+FP}$$
where *G* is the ground truth and *P* is predicted.

The IOU is the area of overlap between the predicted segmentation and the ground truth divided by the area of the union between them.
(18)$$IOU\left(P,G\right)=\frac{\left|P\cap G\right|}{\left|P\cup G\right|}=\frac{TP}{TP+FN+FP}$$ Both the Dice score and the IOU score measure the overlap between the ground truth and the class predicted by the model. Both metrics are always positively correlated. The Dice score is closer to the average performance of the segmentation model, whereas the IoU score represents the worst-case performance of the segmentation model by penalizing the bad classification more.

## 4. Datasets

The datasets used to train and evaluate the SAA-UNet model for CT images were published by the European Institute for Biomedical Imaging Research (EIBIR) [24]. Table 1 shows the details of the limited CT datasets for identifying and quantifying the damage caused by COVID-19 in the lungs. The CT scan of one patient contains a set of slices taken simultaneously from different angles; each slice carries specific information about the lung and the damage of infection to it.

The **MedSeg dataset [60]** contains 100 slices of CT images from more than 40 patients with COVID-19 converted to the Neuroimaging Informatics Technology Initiative format (NIfTI) and is openly accessible from the Italian Society of Radiology (SIRM) [61]. This dataset was segmented by a radiologist using three labels: ground-glass (GGO), consolidation, and pleural effusion, but because of the rarity of pleural effusion, they deleted it and also added lungs and background masks to this dataset [62]. The **Radiopaedia9P dataset [60]** includes whole volumes for nine patients’ CT scans with 829 slices collected from countries across the globe. It includes positive and negative slices, where the radiologist evaluated 373 out of 829 as positive and segmented. This dataset was converted, annotated, and normalized similarly to the MedSeg dataset [62].

The **Zenodo 20P dataset [63]** contains the CT scans with 3520 slices of 20 patients infected with COVID-19 collected from countries across the globe. Two radiologists annotated the left lung, right lung, and infection, then this was verified by an experienced radiologist.

## 5. Data Analysis and Preprocessing

The Hounsfield unit scales the clarity of the CT scan dataset before entering it into the model. The CT scan contrast is different from one dataset to another. This section analyzes and shows the preprocessing of the MedSeg and Radiopaedia 9P datasets, then the Zenodo 20P dataset. Since the MedSeg and Radiopaedia 9P datasets were preprocessed similarly, they can be combined as one dataset to have a more extensive dataset.

### 5.1. Preprocessing of MedSeg and Radiopaedia 9P Datasets

At the start, the slices and related masks were rotated 90 degrees. The radiologists annotated the mask classes for slices into four classes: ground-glass opacity (GGO), consolidation, lungs, and background. The Hounsfield unit (HU) histogram of the MedSeg dataset (100 slices) shows the intensity of pixels confined between −1606 and 597, shown in Figure 6A. The Hounsfield unit (HU) histogram of the Radiopaedia 9P dataset (829 slices) shows the intensity of pixels confined between −1414 and 291, shown in Figure 6C. These datasets have insufficient contrast, so the enhanced contrast method was used. After enhancing the contrast, the HU was normalized and confined between 0 and 1, as shown in Figure 6B for MedSeg and (D) for Radiopaedia 9P. The last row shows the HU before and after preprocessing for combining the two datasets used with the same preprocessing in Figure 6E,F. Two examples of the CT images before and after using the enhanced contrast method are illustrated in Figure 7. The MedSeg dataset has a 512 × 512 resolution (this dataset was not resized because it is limited). The Radiopaedia 9P dataset and the combination of MedSeg and Radiopaedia 9P also have a 512 × 512 resolution resized by shrinking the slices to 128 × 128 using the OpenCV [54] library with inter-area interpolation to decrease the cost compensation. The inter-area interpolation is calculated based on the ratio to shrink the image:(19)$$Ratio=\frac{Imagesiz_{Actual}}{Imagesiz_{New}}$$

The inter-area interpolation ratio is calculated by resizing this dataset with one channel. This ratio is the number of pixels needed to take their average and give it to one pixel.

In binary segmentation, the GGO and consolidation categories were combined as an infection category because the infection segment from the lung regions was our interest.

### 5.2. Preprocessing of Zenodo 20P Dataset

This dataset is in NIfTI format, so first, we extracted the slices from the CT scan and rotated them 90 degrees. These data contain four classes for multi-class segmentation: infection, left lung, right lung, and background. Consequently, the infection segmentation from the slices was our interest in binary segmentation. As shown in Figure 8A, the HU was confined between 4564 and −1023, where the intensity is more in −1000 for air and 0 for water. We normalized the slices directly between 0 and 1. The HU after normalization is shown in Figure 8B. Some slices had a 512 × 512 resolution and the rest 630 × 630, where one patient scan was 401 × 630. We resized the slices to 128 × 128 using the OpenCV library to decrease the cost computation with inter-area interpolation as the other datasets.

## 6. Experiments and Results

To demonstrate the impact of the proposed model, spatial attention and attention UNet (SAA-UNet), we trained it on the four above-mentioned datasets to segment the region of interest (ROI) of damage caused by COVID-19. We trained the model to diagnose whether there was an infection or not. If the infection is present, the model should segment the infection regions. In all experiments, we used all slices of the CT scans, where the CT scans took shots from different angles, and some angles of the slices were taken of a lung region close to other organs.

This section includes the implementation details in Section 6.1, the binary class segmentation experiments in Section 6.2, and the multi-class segmentation experiments in Section 6.3.

### 6.1. Implementation Details

The SAA-UNet model was trained from scratch and implemented with Python 3.8.10, Tensorflow Version 2.9.2, karas 2.9.0, and Google Colab pro+ with GPU. First, we split the datasets into 90% for training and 10% for the testing set, following Ziyang Wang and Irina Voiculescu [34]. After that, we trained the proposed model on a training set with 10-fold cross-validation. K-fold cross-validation is necessary to evaluate the robustness and sensitivity analysis of the proposed model. Hence, ten folds were used to validate the model, and the ten models were trained on the ten validation datasets. Furthermore, these ten trained models were tested on a testing dataset (10% hold-out testing dataset). Table 1 shows the number of slices in each training fold, validation fold, and test set. We used no data augmentation method, like Hoel Kervadec et al. [58]. The Adam optimizer [64] was used, and the learning rate was 10^−4^. The batch size was set to two, and the training epoch for each fold was 150. The hyperparameters used are shown in Table 2.

### 6.2. Binary Class Classification

This type of segmentation was performed to detect the infection on the CT image and also to extract the region of the infection. Firstly, ten-fold cross-validation was performed, and Table 3 shows the experimental results of the SAA-UNet model in the binary class segmentation. All performance metrics are presented as the mean and standard deviation on the validation dataset of the ten models, and Table 4 shows the mean and standard deviation of the ten models obtained from the 10-fold cross-validation experiment on the testing set.

The binary class results of the ten-fold cross-validation in Table 3 show that the SAA-UNet model had the best results on the Radiopaedia 9P dataset and the Zenodo 20P dataset. The mean Dice scores were 0.945 and 0.951 for the Radiopaedia 9P and Zenodo 20P datasets, respectively. The Dice score for the infection region was highest for the Zenodo 20P data, showing a better infection area segmentation. The mean Dice score for the MedSeg dataset was low (0.854) compared to the above-mentioned datasets. The Dice score for the infection class (0.752) was lower than the background class (0.983). If the combination of the MedSeg and Radiopaedia 9P datasets was used for the training, better results were obtained regarding the overall mean Dice score and individual Dice score for both classes. The IOU score defines the ratio of the area of overlap between the predicted segmentation and the ground truth divided by the area of the union between them. The same trend in the IOU score can be observed for all the datasets above. Table 4 summarizes the binary classification results on the testing dataset. The mean Dice and IOU scores were reduced slightly for all the datasets as compared to the results reported on the validation datasets. Some sample predicted slices of the four datasets with COVID-19 infection pixels are illustrated in Figure 9. The confusion matrix of each dataset test set for the SAA-UNet model appears in Figure 10. It shows that SAA-UNet had the closest prediction to the ground truth when predicting images.

Figure 11 illustrates that the SAA-UNet model can predict different sizes and shapes of the infection. Likewise, SAA-UNet can detect if there is no infection in the CT slices with a mean Dice score of 0.99. As a result, SAA-UNet can diagnose and detect the infection effectively even at the beginning of the appearance of pneumonia in the patient’s lung.

In CT imaging, contrast enhancement methods can be applied for infection segmentation to improve the visibility of the areas affected by the infection. This can highlight the areas of interest and make them more distinguishable from the background. However, ensuring that the contrast enhancement does not introduce artifacts or noise that may negatively impact the segmentation accuracy is also important.

The contrast enhancement method was influential in training unclear CT scan images. As shown in Table 5, this method affected the segmentation of the infection and background classes and improved the segmentation process. The MedSeg dataset was improved by 2.4%, whereas Radiopaedia 9P was improved by 1.3% with respect to the mean Dice score compared to without the enhanced contrast method. Furthermore, the mean IOU of MedSeg was improved by 2.2%, whereas the Radiopaedia 9P was the same. This contrast enhancement method of the poor contrast in the MedSeg dataset significantly improved all the evaluation metrics. In addition, the Radiopaedia 9P dataset was improved in the sensitivity and the mean Dice score, especially the Dice score for infection, positively affecting recognizing the foreground and distinguishing it from the background by 1.3%. Figure 12 shows an example of a predicted slice of the same slice fold trained with and without the contrast enhancement method. It is easy to notice that the improvement happened after enhancing the contrast of the MedSeg dataset, whereas for Radiopaedia-9P, the mask was predicted almost as well as without contrast enhancement with no adverse effect on it. As a generalization of the training model, we tested each SAA-UNet trained on one dataset and tested on different datasets. The results are shown in Table 6. The performance metrics decreased while testing on other datasets compared to testing on the same dataset. First, the model trained on Radiopaedia had good results while testing on the MedSeg dataset than on the Zenodo 20P dataset. The SAA-UNet model trained on the Zenodo 20P dataset had the best generalization when tested on the other datasets after applying the contrast enhancement method to ensure the effectiveness of this method. The number of CT slices on which the model was trained and the apparent contrast of the original images led to this generalization and gave promising results. The model trained on the MedSeg dataset showed better results when tested on Radiopaedia 9P than the Zenodo 20P dataset. In contrast, the models trained on the MedSeg + Radiopaedia 9P dataset and then tested on the Zenodo 20P dataset were better than those trained on MedSeg. SAA-UNet had a good generalization for the different training dataset experiments.

### 6.3. Multi-Class Classification

In the multi-class classification experiment, we explored the use of SAA-Unet on many classes, including the lung region and infection or different types of infections. Five classes were considered: background, lungs, infection, consolidation (type of infection), and GGO (type of infection). GGO is a condition in which air is displaced by fluid in the lungs and visible in the CT images as an area of increased attenuation. If a region of normally compressible lung tissue is filled with liquid, it is called pulmonary consolidation. GGO is described as an increase in density with visible blood vessels, whereas the consolidation condition is an increase in the parenchyma density, which conceals the blood vessels. The classification results were obtained on the available classes of the three datasets. The MedSeg dataset had four classes, Radiopaedia 9P four classes, and the Zenodo 20P dataset only three classes. Table 7 illustrates the results of the proposed model in multi-class segmentation as the mean and standard deviation of the validation of the ten models. The mean Dice score was highest for the Zenodo 20P dataset (0.94) and lowest for the MedSeg dataset (0.685). There was only one infection class in the Zenodo 20P dataset, not explaining the type of infection. The infection was also identified in the other two datasets (GGO or consolidation). The mean Dice score for the Radiopaedia 9P dataset was higher than the MedSeg dataset. When combined, the mean Dice score was better than the MedSeg dataset. The other performance metrics also showed the same trend for the three datasets. Table 8 shows the mean and standard deviation of the ten models from the 10-fold cross-validation on the testing dataset. All the performance metrics decreased slightly on the testing dataset, but the trend remained the same. It can be seen from Table 7 and Table 8 that the segmentation of the infection (GGO and Con classes) had a lower Dice score compared to the lungs and background classes. This is because the labeled data for the infection classes were much fewer than for the lung and background classes. Segmenting smaller infection areas with diffused boundaries is challenging, whereas segmenting more significant infection areas with clear boundaries and good contrast is easier. It is evident from Table 7 that increasing the number of labeled slices improved the Dice score of the infection classes in the case of the combination of the MedSeg and Radiopaedia P9 datasets.

Some sample predicted slices of the three datasets with COVID-19 infection pixels are illustrated in Figure 13. It shows that SAA-UNet had a good prediction of the ground truth. The confusion matrix of each dataset for the SAA-UNet model appears in Figure 14. For the MedSeg dataset, both the GGO and Con classes were confused with the lung class (25% and 11%, respectively). The same trend to a lesser extent was observed for the Radiopaedia 9P dataset as well.

Furthermore, we tried different split ratios of the datasets as 80%, 20%, and 70%, 30% for the training and testing set, respectively. Figure 15 shows the average Dice score and IOU for the different split ratios in all four datasets. Different split ratios were tried for both binary and multi-class classification. Figure 15 shows that the effect of different split ratios was insignificant. Both the Dice score and IOU were not degraded significantly when the training ratio was decreased from 90% to 70%. A slight decrease in the Dice score and IOU suggested that more training data improved the segmentation results.

## 7. Discussion

Our proposed method SAA-UNet showed good performance on various datasets related to COVID-19 pneumonia. SAA-UNet also showed better generalization when trained on one dataset and tested on the other datasets. This showed the generalization ability of the SAA-UNet model. The weaker segmentation of the infection classes was also due to the variety of GGO and pulmonary consolidation morphologies. Moreover, a smaller number of pixels classified incorrectly in the image segmentation significantly impacted the Dice coefficient score and IOU [65]. A comparative analysis of different methods used to segment COVID-19 infection in the lung from CT slices is shown in Table 9. In the case of the MedSeg dataset, binary class segmentation, SAA-UNet had better results for the mean Dice score than the other reported results. In the case of the Radiopaedia P9 dataset, our proposed method also outperformed other reported methods, showing a Dice score of 0.94 for binary classification and 0.897 for multi-class classification. The Dice score of SAA-UNet was the best in the case of the Zenodo 20P dataset. Our proposed model performed equally well in binary and multi-class classification (0.95 for binary and 0.94 for multi-class classification). This was due to the Zenodo dataset (2851 slices) containing a higher number of slices available for training as compared to the MedSeg (81 slices) and Radiopaedia P9 (671 slices) datasets in Table 1. Combining the MedSeg and Radiopaedia P9 datasets containing more than 49 subjects produced good binary and multi-class classification results. This showed the efficacy of our proposed model for various datasets in binary and multi-class classification. After comparing to other published results, the SAA-UNet method performed better in the region of interest (ROI) segmentation. It can quantify the severity of the infection and the patient’s condition. Therefore, it can be one of the best methods to be used by the doctor in a follow-up study of the patient. As is evident from Table 5, enhancing the contrast of the CT scan improved the classifier’s performance for better segmentation of the infection. Shiri et al. [66] showed a high overall Dice score (0.98 for lungs and 0.91 for lesions) on a dataset of volumetric CT exams to classify the lungs and pneumonia lesions. Our paper provides more rigorous testing of the model in binary and multi-class classifications. To further prove the generalization ability of our model, we trained the model on one dataset and tested it on the rest of the datasets. The source code of the model is available upon request.

## 8. Conclusions

Diagnosing diseases using computer-aided diagnostic (CAD) methods improves the detection of diseases in real-time. The proposed method, spatial attention and attention mechanism UNet (SAA-UNet), was based on spatial attention UNet (SA-UNet) and attention UNet (Att-UNet) to deal with the challenging structures of COVID-19 pneumonia. SAA-UNet can focus on the foreground to extract the lesion from computed tomography slices. Moreover, the training was optimized by the combination of weighted category cross-entropy loss, Dice loss, and boundary loss, which is useful to extract the hazy edges of the infection and deal with highly imbalanced datasets. The efficacy of the proposed model was established by testing on various datasets, including a smaller number of slices (MedSeg) and more patients (Radiopaedia P9). The performance of the SAA-UNet model was also compared with other reported models. In future work, we will optimize this model further to a larger number of infections in MRI, CT scan, or X-ray images. 

## Figures and Tables

**Figure 1 diagnostics-13-01658-f001:**
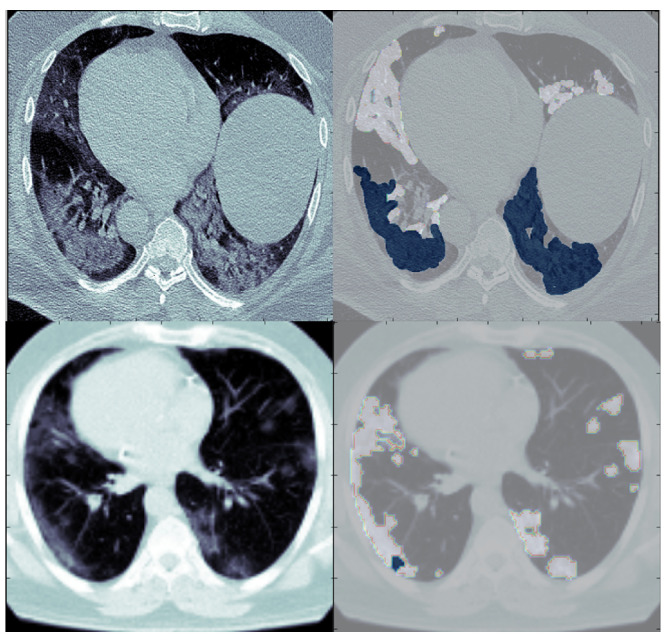
Computed tomography for COVID-19 patients. The ground-glass opacity (GGO) appears in blue and consolidation in white.

**Figure 2 diagnostics-13-01658-f002:**
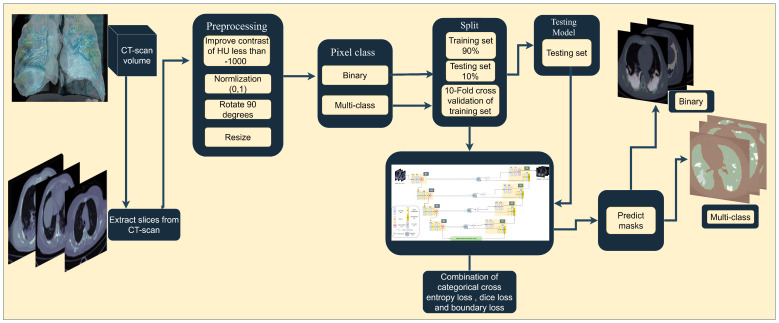
Block diagram of proposed framework for COVID-19 CT segmentation.

**Figure 3 diagnostics-13-01658-f003:**
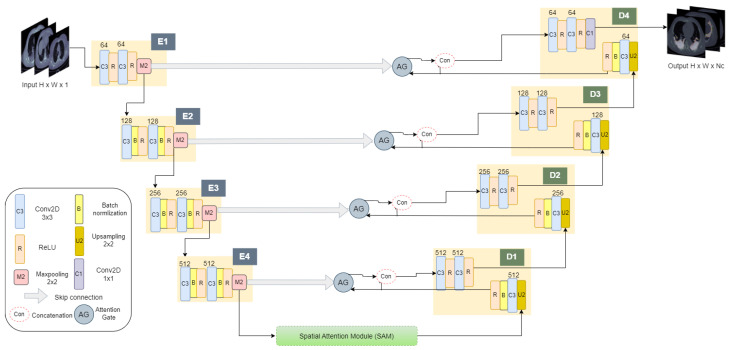
Spatial attention and attention UNet model architecture.

**Figure 4 diagnostics-13-01658-f004:**
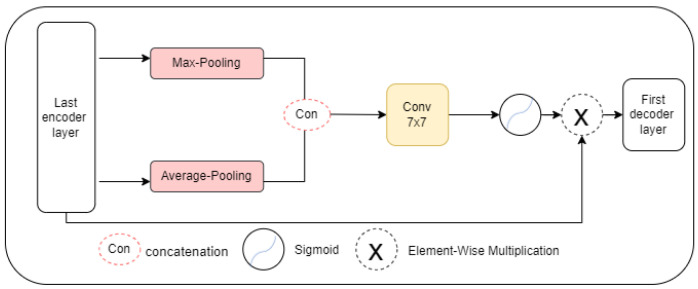
Spatial attention module (SAM) architecture.

**Figure 5 diagnostics-13-01658-f005:**
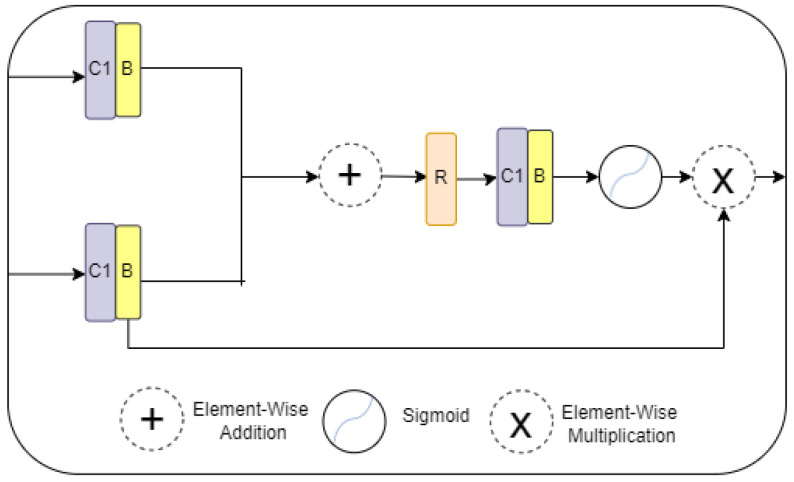
Attention gate (AG) architecture.

**Figure 6 diagnostics-13-01658-f006:**
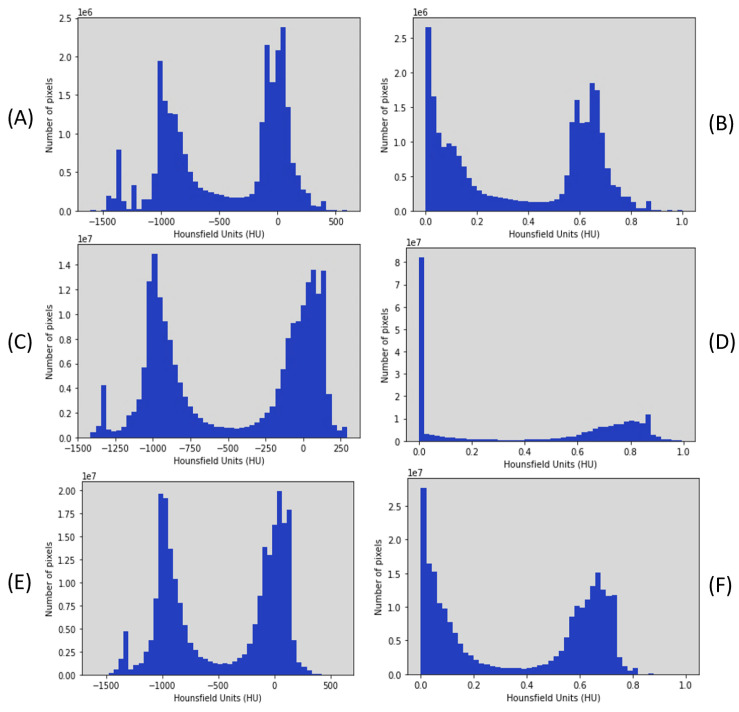
Hounsfield unit (HU) before enhancing the contrast and normalized. (**A**) HU of MedSeg before contrast enhancement, (**B**) HU of MedSeg after enhancement, (**C**) HU of Radiopaedia 9P before enhancement, (**D**) HU of Radiopaedia 9P after enhancement, and (**E**) HU of the combination of them before and (**F**) after the enhancement.

**Figure 7 diagnostics-13-01658-f007:**
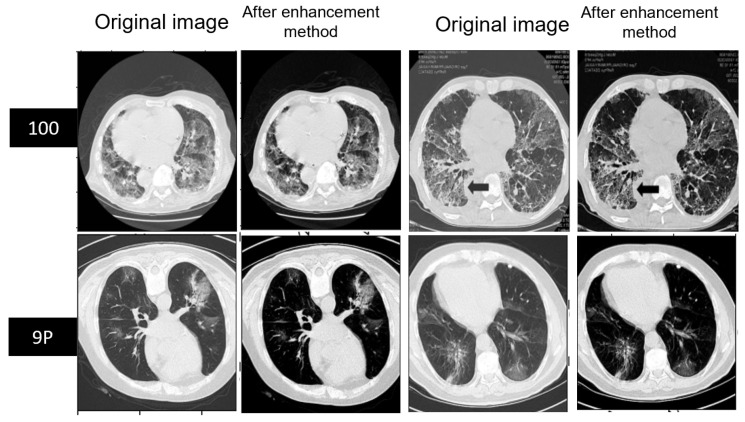
Example of MedSeg and Radiopaedia before and after enhancing the contrast of slices.

**Figure 8 diagnostics-13-01658-f008:**
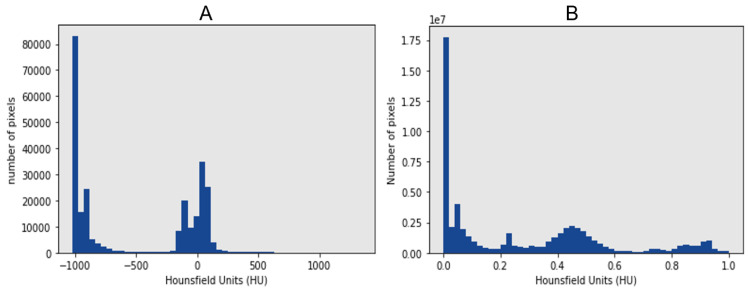
Hounsfield unit (HU) of Zenodo 20P dataset. (**A**) HU before Normalization, (**B**) HU after Normalization.

**Figure 9 diagnostics-13-01658-f009:**
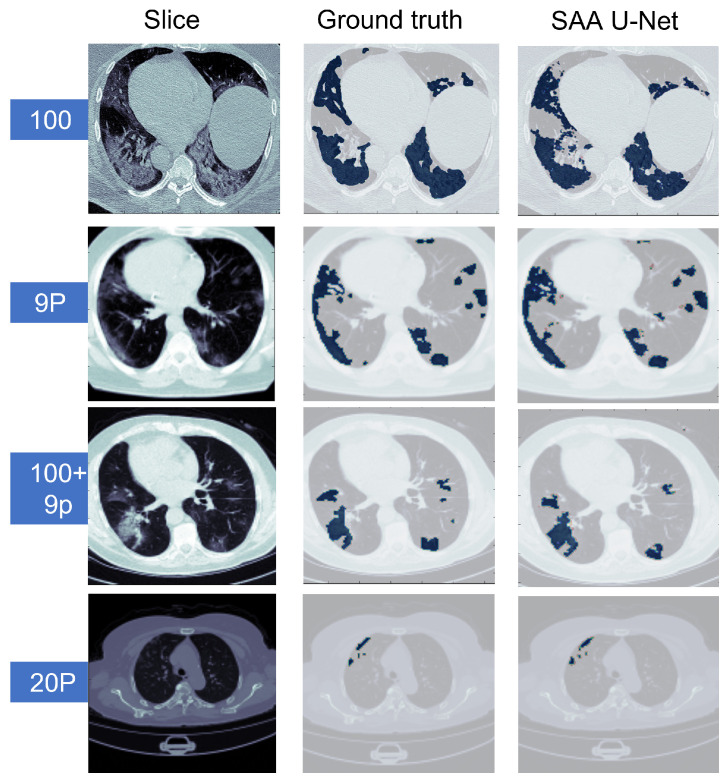
The predicted CT slices of the best fold for each dataset in binary segmentation. The blue color is for infection, and the others are for the background.

**Figure 10 diagnostics-13-01658-f010:**
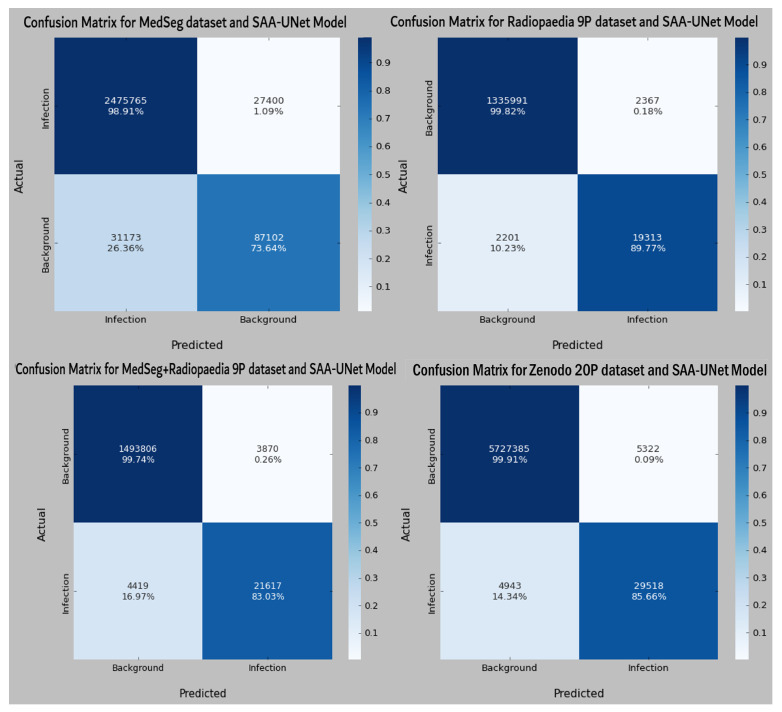
The confusion matrix of each dataset with binary classification for the best fold model from each experiment.

**Figure 11 diagnostics-13-01658-f011:**
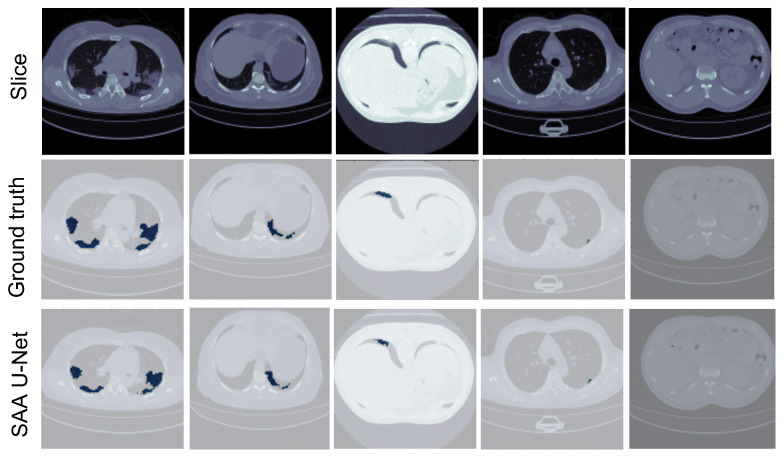
The predicted CT slices for COVID-19 infection with different sizes. The blue color is for infection, and the others are for the background.

**Figure 12 diagnostics-13-01658-f012:**
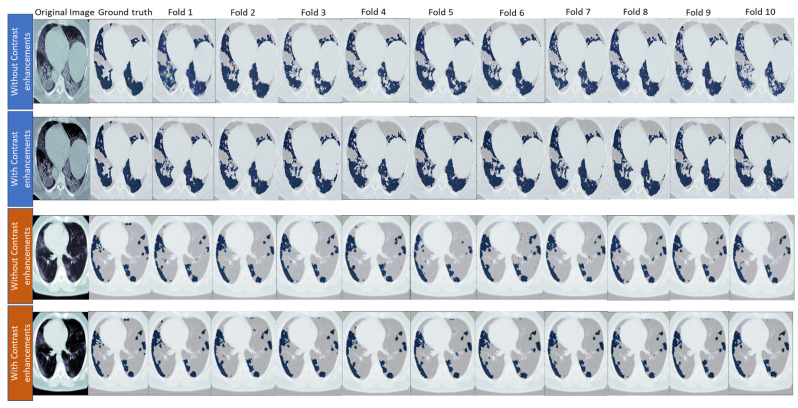
The effect of the contrast enhancement method on predicted images of each fold. The blue in the first two rows is for the MedSeg dataset, and the brown is for the Radiopaedia 9P datasets.

**Figure 13 diagnostics-13-01658-f013:**
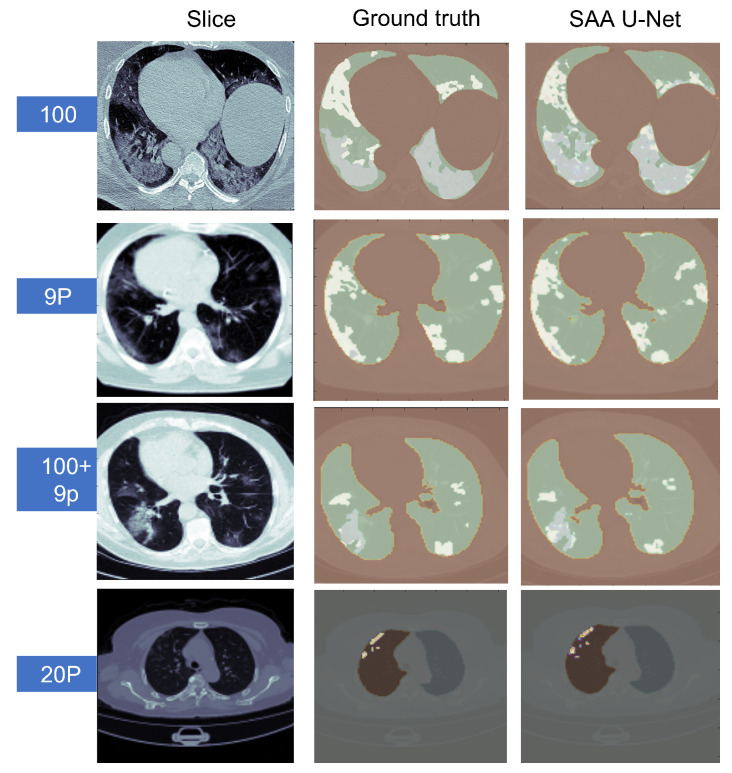
The predicted CT slices of the best fold for each dataset in multi-class segmentation. In the first three datasets, white is for GGO, grey is for consolidation, green is for lungs, and brown is for the background. For the last dataset, white is for infection, brown for the left lung, grey for the right lung, and lighter greyfor the background.

**Figure 14 diagnostics-13-01658-f014:**
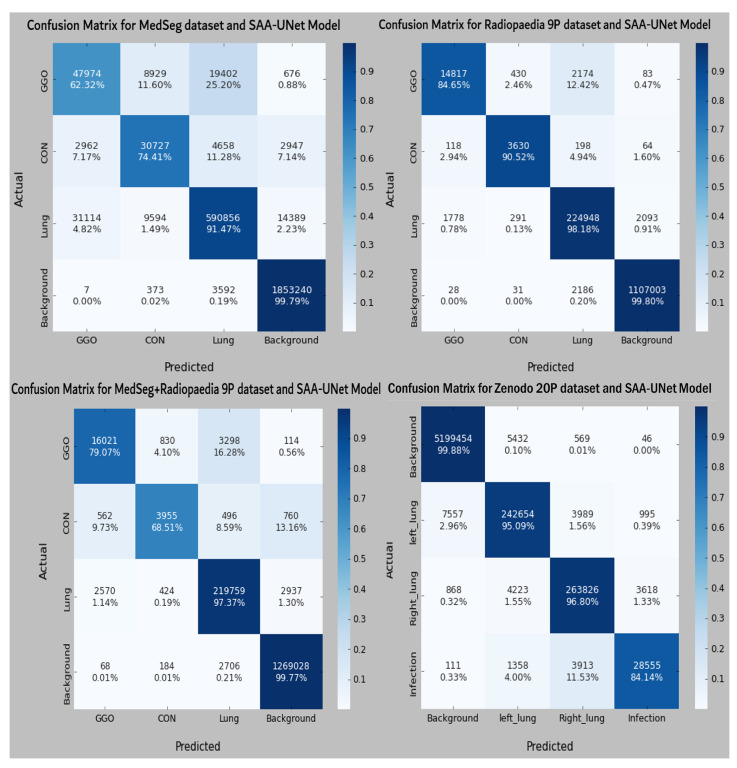
The confusion matrix of all datasets with multi-class segmentation for best fold model from each experiment.

**Figure 15 diagnostics-13-01658-f015:**
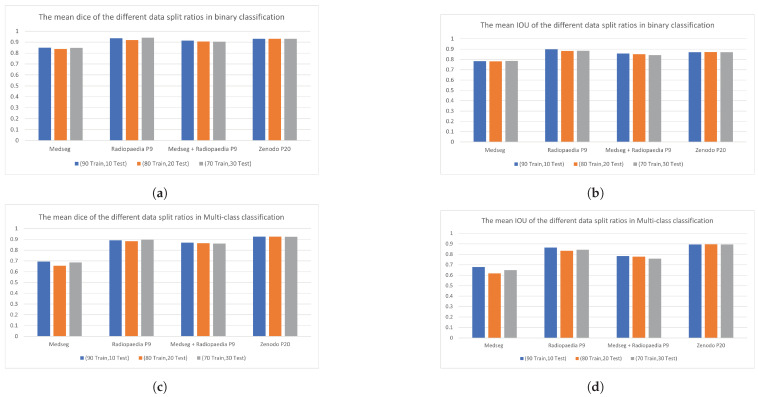
Average Dice score and IOU for different split ratios in binary and multi-class classification. (**a**) Average Dice score in binary classification; (**b**) Average IOU in binary classification; (**c**) Average Dice score in multi-class classification; (**d**) Average IOU in multi-class classification.

**Table 1 diagnostics-13-01658-t001:** Description of the datasets.

Dataset	Patients CT Cases	# of Slices	COVID-19 Infection	Non- COVID-19	Annotation	Training Slices in Each Fold	Validation Slices in Each Fold	Testing
MedSeg	>40	100	96	4	GGO, Consolidation, Lungs, Background	81	9	10
Radiopaedia 9P	9	829	373	456	GGO, Consolidation, Lungs, Background	671	75	83
MedSeg+Radiopaedia9P	>49	929	469	460	GGO, Consolidation, Lungs, Background	752	84	93
Zenodo 20P	20	3520	1793	1727	Infection, Left Lung, Right Lung, Background	2851	317	352

**Table 2 diagnostics-13-01658-t002:** Parameter settings for the training of SAA-UNet model.

Parameter Name	Parameter Value
Number of parameters	18,713,274
Optimizer	Adam
Learning rate	10^−4^
Batch size	2
Epoch	150
Image size	512 × 512, 128 × 128
Data augmentation method	Without

**Table 3 diagnostics-13-01658-t003:** Binary experiments’ results obtained from ten-fold cross-validation experiment (mean ± std).

Dataset	Mean Dice	Dice Inf	Dice Back	Mean IOU	Accuracy	Specificity	Sensitivity	Precision	F1-Score
MedSeg	0.854 ± 0.13	0.725 ± 0.04	0.983 ± 0.003	0.803 ± 0.4	0.970 ± 0.005	0.968 ± 0.005	0.872 ± 0.03	0.892 ± 0.03	0.880 ± 0.03
Radiopaedia 9P	0.945 ± 0.05	0.892 ± 0.01	0.999 ± 0.0002	0.891 ± 0.01	0.997 ± 0.0006	0.997 ± 0.0004	0.934 ± 0.01	0.943 ± 0.007	0.938 ± 0.008
MedSeg + Radiopaedia 9P	0.917 ± 0.08	0.837 ± 0.03	0.997 ± 0.0009	0.849 ± 0.03	0.994 ± 0.001	0.994 ± 0.002	0.904 ± 0.03	0.92 ± 0.02	0.911 ± 0.02
Zenodo 20P	0.951 ± 0.05	0.902 ± 0.05	0.999 ± 0.0004	0.894 ± 0.05	0.998 ± 0.0008	0.998 ± 0.0008	0.935 ± 0.03	0.944 ± 0.03	0.939 ± 0.03

**Table 4 diagnostics-13-01658-t004:** Binary classification results obtained from testing the ten trained models on the testing dataset (mean ± std).

Dataset	Mean Dice	Dice Inf	Dice Back	Mean IOU	Accuracy	Specificity	Sensitivity	Precision	F1-Score
MedSeg	0.848 ± 0.14	0.708 ± 0.01	0.988 ± 0.0004	0.783 ± 0.005	0.978 ± 0.0008	0.976 ± 0.0009	0.858 ± 0.01	0.872 ± 0.01	0.865 ± 0.004
Radiopaedia 9P	0.936 ± 0.06	0.875 ± 0.009	0.998 ± 5.5 × 10^−5^	0.899 ± 0.003	0.997 ± 0.0001	0.996 ± 9.38 × 10^−5^	0.940 ± 0.005	0.948 ± 0.003	0.943 ± 0.002
MedSeg + Radiopaedia 9P	0.914 ± 0.08	0.831 ± 0.03	0.997 ± 6.1 × 10^−5^	0.857 ± 0.003	0.995 ± 0.0001	0.995 ± 0.0002	0.911 ± 0.005	0.924 ± 0.005	0.917 ± 0.002
Zenodo 20P	0.93 ± 0.07	0.861 ± 0.007	0.999 ± 1.6 × 10${}^{-5}$	0.87 ± 0.002	0.998 ± 3.3 × 10${}^{-5}$	0.998 ± 4.1 × 10${}^{-5}$	0.921 ± 0.005	0.930 ± 0.005	0.926 ± 0.001

**Table 5 diagnostics-13-01658-t005:** The effect of the contrast enhancement on the MedSeg and Radiopaedia 9P datasets.(bold text for the best).

Contrast Enhancement	Mean Dice	Dice Inf	Dice Back	Mean IOU	Accuracy	Specificity	Sensitivity	Precision	F1-Score
MedSeg (without)	0.824 ± 0.16	0.66± 0.07	0.987 ± 0.001	0.761 ± 0.04	0.976 ± 0.003	0.974 ± 0.006	0.839 ± 0.03	0.865 ± 0.06	0.846 ± 0.03
**MedSeg** **(with)**	**0.848 ±** **0.14**	**0.708 ±** **0.01**	**0.988 ±** **0.0004**	**0.783 ±** **0.005**	**0.978 ±** **0.0008**	**0.976 ±** **0.0009**	**0.858 ±** **0.01**	**0.872 ±** **0.01**	**0.865 ±** **0.004**
Radiopaedia 9P (without)	0.923 ± 0.1	0.847 ± 0.04	0.998 ± 0.0003	0.899 ± 0.01	0.997 ± 0.0004	0.996 ± 0.0008	0.939 ± 0.01	0.948 ± 0.005	0.943 ± 0.007
**Radiopaedia 9P** **(with)**	**0.936 ±** **0.06**	**0.875 ±** **0.009**	0.998 ± 5.5 × 10${}^{-5}$	0.899 ± 0.003	0.997 ± 0.0001	0.996 ± 9.38 × 10${}^{-5}$	**0.940 ±** **0.005**	0.948 ± 0.003	0.943 ± 0.002

**Table 6 diagnostics-13-01658-t006:** Training SAA-UNet on one dataset and testing it on different datasets.

Trained Dataset	Tested Dataset	Mean Dice	Dice Inf	Dice Back	Mean IOU	Accuracy	Specificity	Sensitivity	Precision	F1-Score
Radiopaedia 9P	MedSeg	0.71 ± 0.2	0.462 ± 0.02	0.957 ± 0.009	0.645 ± 0.02	0.920 ± 0.0003	0.922 ± 0.02	0.773 ± 0.03	0.731 ± 0.007	0.744 ± 0.02
Radiopaedia 9P	Zenodo 20P	0.579 ± 0.4	0.172 ± 0.04	0.977 ± 0.007	0.522 ± 0.01	0.957 ± 0.01	0.956 ± 0.01	0.78 ± 0.03	0.545 ± 0.002	0.567 ± 0.01
Zenodo 20P	MedSeg	0.770 ± 0.2	0.566 ± 0.03	0.973 ± 0.002	0.686 ± 0.02	0.957 ± 0.002	0.952 ± 0.003	0.724 ± 0.02	0.908 ± 0.003	0.783 ± 0.02
Zenodo 20P	Radiopaedia 9P	0.802 ± 0.2	0.609 ± 0.03	0.996 ± 0.0002	0.607 ± 0.01	0.992 ± 0.0004	0.991 ± 0.0004	0.782 ± 0.01	0.730 ± 0.01	0.657 ± 0.02
Zenodo 20P	MedSeg + Radiopaedia 9P	0.809 ± 0.2	0.624 ± 0.02	0.994 ± 0.0003	0.619 ± 0.01	0.988 ± 0.0006	0.988 ± 0.0006	0.80 ± 0.02	0.735 ± 0.008	0.673 ± 0.01
MedSeg	Radiopaedia 9P	0.71 ± 0.3	0.424 ± 0.06	0.996 ± 0.0002	0.649 ± 0.01	0.993 ± 0.0003	0.993 ± 0.0003	0.859 ± 0.02	0.719 ± 0.007	0.701 ± 0.01
MedSeg	Zenodo 20P	0.441 ± 0.4	0.04 ± 0.002	0.839 ± 0.02	0.371 ± 0.02	0.724 ± 0.03	0.721 ± 0.03	0.646 ± 0.02	0.505 ± 0.001	0.436 ± 0.01
MedSeg + Radiopaedia 9P	Zenodo 20P	0.57 ± 0.4	0.169 ± 0.04	0.971 ± 0.008	0.507 ± 0.014	0.945 ± 0.014	0.943 ± 0.012	0.785 ± 0.02	0.535 ± 0.008	0.549 ± 0.02

**Table 7 diagnostics-13-01658-t007:** Multi-class experiments results obtained from test ten models on the validation set (mean ± std).

Dataset	Mean Dice	Dice GGO	Dice Con	Dice Inf	Dice Back	Dice Lung	Mean IOU	Accuracy	Specificity	Sensitivity	Precision	F1-Score
MedSeg	0.685 ± 0.25	0.530 ± 0.07	0.367 ± 0.07	-	0.992 ± 0.003	0.85 ± 0.04	0.659 ± 0.03	0.952 ± 0.009	0.984 ± 0.003	0.759 ± 0.05	0.785 ± 0.03	0.762 ± 0.03
Radiopaedia 9P	0.897 ± 0.07	0.794 ± 0.04	0.871 ± 0.04	-	0.998 ± 0.0001	0.926 ± 0.02	0.839 ± 0.02	0.994 ± 0.0007	0.998 ± 0.0001	0.894 ± 0.02	0.917 ± 0.02	0.904 ± 0.01
MedSeg + Radiopaedia 9P	0.873 ± 0.1	0.751 ± 0.04	0.81 ± 0.03	-	0.997 ± 0.0004	0.933 ± 0.01	0.775 ± 0.03	0.989 ± 0.002	0.996 ± 0.0006	0.846 ± 0.02	0.875 ± 0.02	0.855 ± 0.02
Zenodo 20P	0.940 ± 0.04	-	-	0.880 ± 0.05	0.999 ± 0.0005	L = 0.952 ± 0.02 R = 0.931 ± 0.03	0.909 ± 0.04	0.995 ± 0.002	0.998 ± 0.0007	0.946 ± 0.02	0.953 ± 0.02	0.949 ± 0.02

**Table 8 diagnostics-13-01658-t008:** Multi-class experiments results obtained from test ten models on the test set (mean ± std).

Dataset	Mean Dice	Dice GGO	Dice Con	Dice Inf	Dice Back	Dice Lung	Mean IOU	Accuracy	Specificity	Sensitivity	Precision	F1-Score
MedSeg	0.693 ± 0.27	0.557 ± 0.05	0.311 ± 0.03	-	0.993 ± 0.001	0.909 ± 0.08	0.679 ± 0.02	0.964 ± 0.002	0.988 ± 0.0008	0.775 ± 0.03	0.795 ± 0.02	0.78 ± 0.02
Radiopaedia 9P	0.891 ± 0.09	0.768 ± 0.03	0.859 ± 0.02	-	0.997 ± 0.0001	0.941 ± 0.01	0.865 ± 0.007	0.993 ± 0.0003	0.998 ± 8.95 × 10${}^{-5}$	0.917 ± 0.01	0.931 ± 0.01	0.923 ± 0.005
MedSeg + Radiopaedia 9P	0.870 ± 0.10	0.752 ± 0.02	0.794 ± 0.02	-	0.997 ± 6.5 × 10${}^{-5}$	0.937 ± 0.008	0.783 ± 0.003	0.99 ± 0.0001	0.997 ± 4.5 × 10${}^{-5}$	0.854 ± 0.004	0.88 ± 0.005	0.864 ± 0.002
Zenodo 20P	0.926 ± 0.06	-	-	0.84 ± 0.01	0.998 ± 0.0001	L = 0.945 ± 0.004 R = 0.919 ± 0.004	0.894 ± 0.005	0.994 ± 0.0003	0.998 ± 9.3 × 10${}^{-5}$	0.937 ± 0.005	0.945 ± 0.002	0.941 ± 0.003

**Table 9 diagnostics-13-01658-t009:** Comparison analysis of different segmentation methods in related work with the proposed method. SAA-UNet1 shows the results of ten-fold validation, and SAA-UNet2 shows the result of the testing dataset.

Dataset	Class Classification	Model	Mean Dice	Mean IOU	Accuracy	Specificity	Sensitivity	Precision	F1-Score
MedSeg	Binary	Plug-and-play Attention UNet [33]	0.84	0.74	-	-	-	-	-
	Binary	HADCNet [42]	0.792	-	0.970	0.985	0.871	-	-
	Binary	MiniSeg [44]	0.759	0.822	-	0.977	0.8495	-	-
	Binary	SAA-UNet1	0.854	0.803	0.970	0.968	0.872	0.892	0.88
	Binary	SAA-UNet2	0.85	0.78	0.978	0.976	0.858	0.872	0.865
	Multi-class	SAA-UNet1	0.685	0.659	0.952	0.984	0.759	0.785	0.762
	Multi-class	SAA-UNet2	0.693	0.679	0.964	0.988	0.775	0.795	0.78
Radiopaedia 9P	Binary	MPS-Net [39]	0.83	0.74	-	0.9988	0.8406	-	-
	Binary	DUDA-Net [46]	0.87	0.771	0.991	0.996	0.909	-	-
	Binary	HADCNet [42]	0.796	-	0.991	0.994	0.912	-	-
	Binary	MiniSeg [44]	0.80	0.853	-	0.992	0.906	-	-
	Binary	SAA-UNet1	0.945	0.891	0.997	0.997	0.934	0.943	0.938
	Binary	SAA-UNet2	0.94	0.90	0.997	0.996	0.940	0.948	0.943
	Multi-class	SAA-UNet1	0.897	0.839	0.994	0.998	0.894	0.917	0.904
	Multi-class	SAA-UNet2	0.89	0.87	0.993	0.998	0.917	0.931	0.923
MedSeg+ Radiopaedia 9P	Binary	TV UNet [31]	0.864	0.995	-	-	0.85	0.87	-
	Binary	Channel-attention UNet [30]	0.83	-	-	-	-	-	-
	Binary	Ensemble UNet & majority voting [38]	0.85	-	-	0.994	0.891	-	-
	Binary	ADID-UNet [41]	0.803	-	0.97	0.9966	0.797	0.848	0.82
	Binary	A-SegNet [40]	0.896	-	-	0.995	0.927	-	-
	Binary	SAA-UNet1	0.917	0.849	0.994	0.994	0.904	0.92	0.911
	Binary	SAA-UNet2	0.90	0.84	0.993	0.998	0.917	0.931	0.923
	Multi-class	DDANet [49]	0.78	-	-	0.992	0.884	-	-
	Multi-class	SAA-UNet1	0.873	0.775	0.989	0.996	0.846	0.875	0.855
	Multi-class	SAA-UNet2	0.87	0.78	0.99	0.997	0.854	0.88	0.864
Zenodo 20P	Binary	FCN-8s Light-UNet [27]	-	-	(1) 0.98 (2) 0.98	-	(1) 0.50 (2) 0.57	(1) 0.85 (2) 0.96	(1) 0.57 (2) 0.64
	Binary	3-Encoder, 3Decoder [43]	-	0.799	0.972	0.9499	0.9499	0.993	-
	Binary	LungINFseg [47]	0.803	0.688	0.989	0.995	0.831	-	-
	Binary	contour-enhanced attention decoder CNN [48]	0.88	0.75	-	0.998	0.90	0.856	-
	Binary	Focal attention module with DeepLabV3+ [45]	0.885	-	-	-	-	-	-
	Binary	HADCNet [42]	0.723	-	0.987	0.997	0.694	-	-
	Binary	MiniSeg [44]	0.763	0.845	-	0.991	0.851	-	-
	Binary	SAA-UNet1	0.951	0.894	0.998	0.998	0.935	0.944	0.939
	Binary	SAA-UNet2	0.93	0.88	0.998	0.998	0.921	0.93	0.926
	Multi-class	QAP-Net [34]	-	0.816	0.9976	0.998	0.958	0.846	-
	Multi-class	MultiResUNet [35]	0.88	-	-	-	-	-	-
	Multi-class	SAA-UNet1	0.940	0.909	0.995	0.998	0.946	0.953	0.949
	Multi-class	SAA-UNet2	0.931	0.899	0.994	0.998	0.937	0.945	0.941

## Data Availability

All datasets are publically available.

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
