# Peer review of "SAA-UNet: Spatial Attention and Attention Gate UNet for COVID-19 Pneumonia Segmentation from Computed Tomography"

_diagnostics, 2023, doi:10.3390/diagnostics13091658_

Round 1
Reviewer 1 Report
Authors should check Some Missing references like in the first paragraph in methodology.
Typos should be corrected like in figure 2 caption. The other performance measures such as accuracy and specificity were mentioned in section 5 but their values not highlighted in the Abstract and were not used in the comparison with other related previous work. The pseudo code in figure 6 needs more details. It needs also some explanations such as the stopping condition for example. The pseudo code should not placed in a figure, but in an algorithm table. In figure 8, the authors demonstrate image contrast enhancement. This process affects the pneumonia areas due to their low intensity values. Authors should highlight and discuss such impact and demonstrate how it affects their results with and without the enhancement. The authors mentioned that they split the datasets into 90% for training and 10% for testing. The authors should demonstrate the effect and results of also 80% for training and 20% for testing. What about 70% for training also? What impact will it have on the results? The authors should compare their results with some other similar works that have higher percentages for performance parameters such as: Shiri, Isaac, et al. "COLI‐Net: Deep learning‐assisted fully automated COVID‐19 lung and infection pneumonia lesion detection and segmentation from chest computed tomography images." International journal of imaging systems and technology 32.1 (2022): 12-25. What is the advantage of the proposed work over that previous related work? Spacings and missing references have to be checked.Author Response
Answers to the Comments of Reviewer 1
We want to thank the respected reviewer for his valuable comments. We have revised the paper according to the comments of respected reviewer. Please find below our answers to the comments.
Comment 1: Authors should check Some Missing references like in the first paragraph in methodology.
Answer: Thanks for pointing out the mistake. We have fixed the references.
Comment 2: Typos should be corrected like in figure 2 caption.
Answer: Thanks for pointing out the mistake. We have fixed the typos.
Comment 3: The other performance measures such as accuracy and specificity were mentioned in section 5 but their values not highlighted in the Abstract and were not used in the comparison with other related previous work.
Answer: We have added the other performance measures to the table comparison with related work. Also, We have added accuracy and specificity in the abstract.
Comment 4: The pseudo-code in figure 6 needs more details. It needs also some explanations such as the stopping condition for example. The pseudo code should not placed in a figure, but in an algorithm table.
Answer: We have revised the pseudo-code and placed it in an algorithm table. Please see Table 1 on page 9.
Comment 5: In Figure 8, the authors demonstrate image contrast enhancement. This process affects the pneumonia areas due to their low-intensity values. Authors should highlight and discuss such impact and demonstrate how it affects their results with and without the enhancement.
Answer: We have added a demonstration of the effects of results with and without the enhancement. Lines 505-509 and Lines 515-522
Comment 6: The authors mentioned that they split the datasets into 90% for training and 10% for testing. The authors should demonstrate the effect and results of also 80% for training and 20% for testing. What about 70% for training also? What impact will it have on the results?
Answer: We have added results for different splits. Please see Figure 15 and the explanation from Lines 572-576
Comment 7: The authors should compare their results with some other similar works that have higher percentages for performance parameters such as: Shiri, Isaac, et al. "COLI‐Net: Deep learning‐assisted fully automated COVID‐19 lung and infection pneumonia lesion detection and segmentation from chest computed tomography images." International journal of imaging systems and technology 32.1 (2022): 12-25.
Answer: We have added this reference and described the difference. Our model is tested on more than two classes and different datasets to prove the generalization capability of the model. Please see lines 602-606.
Comment 8: What is the advantage of the proposed work over that previous related work?
Answer: This research is distinguished from previous research by proposing an SAA-UNet algorithm and a framework that can determine the region of interest of Covid-19 infection from tomography images efficiently and effectively, as provided in Table 10.
Comment 9: Spacings and missing references have to be checked.
Answer: Yes, we checked and fixed the spacing references.
Reviewer 2 Report
Dear Authors, I am truly grateful for the opportunity to review your esteemed paper.
The authors present a novel method by incorporating an attention mechanism into the UNet segmentation algorithm and provide comparative analysis results through benchmark tests with various methods.
The proposed method is a fairly typical application study, involving modifications to the UNet model's skip-connections and the final convolutional block. Nevertheless, the authors have compared and analyzed numerous publicly available datasets and models, and the paper reports sufficiently valuable results.
If the minor revisions mentioned above are addressed, it seems to be a good manuscript that is sufficiently publishable.
* SAM & Attention gate
Utilizing the attention mechanism is indeed an excellent choice; however, to enhance its performance, it is not only the attention-augmented features that should be used, but also the values before attention is applied. For example, even Transformers do not merely use attention results but concatenate them with the original values and pass them to the next layer. The authors, however, seem to use only the values after applying attention, and I would like to inquire about the rationale behind this choice.
* Please share the source code.
There are numerous minor corrections needed throughout the paper.
The overall text contains redundant parts, making the sentence structures too lengthy
The background sentence in the Abstract is excessively long. Please shorten the content up to "In this paper, we propose..."
In the Abstract, the sentence "Moreover, it also performed better" is unclear, as it does not specify what it performed better than.
There appears to be an issue with the reference section in line 91, as it contains "[? ]".
The related work section mentions too many methods. Instead of mentioning a wide variety of methods, it would be more appropriate to describe papers introducing key concepts that this study is based on, such as UNet, Attention gate UNet, and spatial attention.
In section 3.1 "Pre-processing of Images," the most fundamental input image size is not mentioned.
Author Response
Answers to the Comments of Reviewer 2
We want to thank the respected reviewer for his valuable comments. We have revised the paper according to the comments of the respected reviewer. Please find below our answers to the comments.
Comments and Suggestions for Authors
Dear Authors, I am truly grateful for the opportunity to review your esteemed paper.
The authors present a novel method by incorporating an attention mechanism into the UNet segmentation algorithm and provide comparative analysis results through benchmark tests with various methods.
The proposed method is a fairly typical application study, involving modifications to the UNet model's skip connections and the final convolutional block. Nevertheless, the authors have compared and analyzed numerous publicly available datasets and models, and the paper reports sufficiently valuable results.
If the minor revisions mentioned above are addressed, it seems to be a good manuscript that is sufficiently publishable.
We are very thankful to the respected reviewer for his support.
Comment 1: * SAM & Attention gate
Utilizing the attention mechanism is indeed an excellent choice; however, to enhance its performance, it is not only the attention-augmented features that should be used, but also the values before attention is applied. For example, even Transformers do not merely use attention results but concatenate them with the original values and pass them to the next layer. The authors, however, seem to use only the values after applying attention, and I would like to inquire about the rationale behind this choice.
Answer: We thank the respected reviewer for his valuable comment. Pneumonia is distributed within the lung region and has fuzzy edges that may mix with the lung tissue or other noise. Moreover, the distribution of infection or types of infection pixels is one of the challenges in this problem. Initially, we started experimenting with Attention U-Net, which gave pretty good results, but some pixels significantly interfered with the lung tissue. The reason is that much unimportant information is transferred from the encoder to the decoder by the bridge. At the same time, we want to focus on spatial information of the region of interest (ROI) areas inside the lung and extract them correctly, especially when the disease is in an early stage with a tiny size, an essential point in the SAA-UNet algorithm.
Comment 2: * Please share the source code.
Answer: We have added a line at Line 606 about the source code. We will provide the source code of the model upon request.
Comment 3: There are numerous minor corrections needed throughout the paper.
-The overall text contains redundant parts, making the sentence structures too lengthy:
The background sentence in the Abstract is excessively long. Please shorten the content up to "In this paper, we propose..."
Answer: Thanks a lot for pointing out the mistakes. We have corrected the errors.
Comment 4: In the Abstract, the sentence "Moreover, it also performed better" is unclear, as it does not specify what it performed better than.
Answer: Thanks a lot. We have updated it.
Comment 5: There appears to be an issue with the reference section in line 91, as it contains "[? ]".
Answer: We updated this reference.
Comment 6: The related work section mentions too many methods. Instead of mentioning a wide variety of methods, it would be more appropriate to describe papers introducing key concepts that this study is based on, such as UNet, Attention gate UNet, and spatial attention.
Answer: We added UNet, Attention gate UNet, and spatial attention UNet to Related work.
Comment 7: In section 3.1, "Pre-processing of Images," the most fundamental input image size is not mentioned
Answer: It is already mentioned for each dataset in section 5.
Round 2
Reviewer 1 Report
Good work with appropriate methodology and results.